# Drug Prescriptions Requiring Compounding at a Canadian University Affiliated Pediatric Hospital: A Cross-Sectional Study

**DOI:** 10.3390/children10010147

**Published:** 2023-01-11

**Authors:** Émilie Kate Landry, Julie Autmizguine, Sophie Bérubé, Raphael Kraus, Marie-Élaine Métras, Denis Lebel, Catherine Litalien

**Affiliations:** 1The Rosalind & Morris Goodman Family Pediatric Formulations Centre of the CHU Sainte-Justine, Montreal, QC H3T 1C5, Canada; 2Department of Pediatrics, CHU Sainte-Justine, Montreal, QC H3T 1C5, Canada; 3Department of Pharmacology and Physiology, Faculty of Medicine, Université de Montréal, Centre-Ville Station, Montreal, QC H3C 3J7, Canada; 4Department of Pharmacy, CHU Sainte-Justine, Montreal, QC H3T 1C5, Canada

**Keywords:** pediatric formulations, compounded drugs, compounding, hospital pharmacy, liquid oral dosage form, solid oral dosage form

## Abstract

Despite ongoing international efforts, many drugs administered to children must be compounded from dosage forms designed for adults because they remain unavailable in commercial formulations that suit their needs. Even though oral drug compounding is common in pediatrics, the extent of this practice has not been well described in recent years. This cross-sectional and retrospective study was conducted at a Canadian university-affiliated, 484-bed, tertiary care pediatric hospital and its rehabilitation centre on two randomly selected days. A total of 606 hospitalized children with 5465 prescriptions were included. Overall, compounded drugs for enteral administration (CDEA) represented 13% of all prescriptions (enteral and parenteral) and 23% of prescriptions for enteral administration. Of the 390 prescribed drugs, 122 required compounding. CDEA were mostly liquids (n = 478 [67%]) and mainly included drugs of the central nervous (35%), cardiovascular (21%), and gastro-intestinal (12%) systems. Nearly half (N = 298 [49%]) of children had at least one CDEA prescribed in their medical file. Many CDEA are available as commercial products in other jurisdictions. Collaboration is needed between all stakeholders to make these drugs available to Canadian children.

## 1. Introduction

Providing optimal pharmacological treatment to children comes with multiples challenges, including administrating drugs in the absence of suitable child-friendly formulations. In such instances, the dosage forms designed for adults need to be altered to meet children’s needs, a practice referred to as compounding. Such manipulations are generally performed by pharmacists or at the point of use by nurses or parents [1]. Compounding is a long-standing practice in pharmacological treatments, particularly for children, as it serves an important unmet medical need for this population [2,3]. However, compounded medications cannot be viewed as equivalent to commercial forms as they fall out of Good Manufacturing Practices [4], and are not overseen by regulatory agencies (such as Health Canada) for efficacy, safety, and quality before they reach patients [5]. This off-label use can lead to sub-optimal adherence due to unpleasant taste, underdosing with resultant therapeutic failure, and overdosing leading to unintended adverse events [5,6,7]. In a recent report on pediatric medication errors in Canadian community pharmacies over a 5-year period, several incidents, with varying levels of harm, stemmed from erroneous concentration and/or dose volume calculations during the preparation of compounded liquid formulations and subsequently during the provision of directions for use [8].

A recent study conducted in a Canadian tertiary pediatric centre showed that almost 50% of most frequently compounded drugs had an available commercial child-friendly drug equivalent in other jurisdictions [9]. Another study showed that 60% of new drugs approved for use by Health Canada in children less than 6 years of age between 2007–2016 were not marketed in a child-friendly formulation [10]. However, to our knowledge, there are no Canadian studies nor data available on the extent of compounding for orally administered drugs in the hospital setting. Healthcare professionals should be aware of this information as compounding, with its inherent deficiencies, can impact the efficacy and safety profile of the drugs they are prescribing daily.

The primary objective of this study was to determine the proportion of prescriptions for compounded drugs for enteral administration (CDEA), among all enteral and parenteral prescriptions, in children under the age of 18 years hospitalized at a Canadian pediatric hospital and its rehabilitation Centre. Secondary objectives were to determine the proportion of prescribed drugs requiring compounding and the proportion of children prescribed at least one CDEA during their hospital stay.

## 2. Materials and Methods

### 2.1. Study Design

This retrospective single-centre cross-sectional study was conducted at the Centre Hospitalier Universitaire (CHU) Sainte-Justine, a university-affiliated, 484-bed, tertiary care pediatric hospital in Montreal, Quebec, Canada. All prescriptions of drugs for enteral and parenteral administration for hospitalized children were identified using the hospital pharmacy database on two randomly selected days between 20 March 2019 and 19 March 2020, purposely one during the summer (23 July 2019) and one during the winter (26 February 2020) to account for seasonal variation on drug prescriptions. Dates were selected using RAND function in Microsoft Excel version 16. This study was approved by the Research Ethics Board of the CHU Sainte-Justine Research Centre on 3 June 2020 (approval code 2021-2876). Informed consent was not required because of the retrospective nature of this study.

### 2.2. Study Population

Children who were hospitalized at the CHU Sainte-Justine as well as its rehabilitation centre during the study dates were included in this study if they were aged between 0 and 18 years and had at least one prescription in their medical file on the selected days. Patients in the emergency department, in the medical day care unit, or in the delivery room, or those hospitalized for an elective one-day surgery, were excluded. A child hospitalized during both study dates was considered as two distinct patients, since one of the initial interests of this study was to obtain a comparison portrait of two separate days. Patients were grouped according to a modified International Council for Harmonization (ICH) pediatric age classification [11]. A cut-off at the age of 8 years was added to better define compounding challenges in patients 2 to 8 years and 8 to 12 years.

### 2.3. Prescriptions of Interest

For the purpose of this study, enteral routes of administration included administration by mouth, by nasogastric tube, by nasojejunal tube, by gastrostomy, by gastrojejunostomy, or by jejunostomy. Intrarectal treatments were not included in enteral prescriptions, as compounding challenges intrinsic to this administration route are different. The American Hospital Formulary Service (AHFS) Pharmacologic–Therapeutic Classification was used for all prescriptions.

CDEA was defined as any drug requiring manipulation from its original packaging to be adequately administered to children. A liquid CDEA was defined as tablets, capsules, injectables, or active pharmaceutical ingredients (API) used with a liquid vehicle to produce a suspension, a solution, or a syrup or as an injectable given orally. All antibiotics manufactured as powder-filled bottles requiring dilution, as indicated in the product monograph, were not considered as CDEA. Solid CDEA included tablet splitting, spreading of API after capsule opening, powder measurement, or API encapsulation. Such manipulations may have been performed either by the hospital pharmacy staff or bed-side nurses. If a manufactured packet of premeasured powder required splitting before dilution for pediatric dosing, then it was considered to be a solid CDEA. This was the case for esomeprazole (Nexium^®^). Polyethylene glycol 3350 (Lax A Day^®^), manufactured in large volumes of powdered API (>500 g) requiring measurement and dilution before administration, was not considered as a CDEA. All prescription definitions used in this study are described in Table 1.

### 2.4. Data Extraction

Patients’ demographic information was extracted from the electronic medical files and from the hospital pharmacy database. The hospital pharmacy database also provided all information regarding drug prescriptions, including the need for manipulation by the pharmacy team, which comprised not only compounding, but also bulk production, and oral syringe preparation. These prescriptions were precisely screened for CDEA according to this study’s definitions. Drug combinations were analyzed separately from their active ingredients’ isolated form (e.g., amoxicillin-clavulanate and amoxicillin). A prescription change in the dose, route, or frequency of administration on the study date was considered to be a new prescription. No screening was performed for parenteral prescriptions, as this was not the focus of this study. The 11th International Classification of Diseases (ICD-11) was used for the diagnosis on admission.

If indicated, the compounding recipes for liquids used by the hospital pharmacy and those provided by the pharmacological treatment software (Rx Vigilance) were used to determine the type of CDEA. If multiple recipes existed for a single drug, the tablet-based liquid was selected, as it would be in the pharmacy practice at the CHU Sainte-Justine. All solid forms with a prescribed dose smaller than the available dosage forms were screened for tablet splitting, capsule opening, or other solid CDEA, as they were not labelled with the requirement for manipulation.

### 2.5. Quality Control

Data entry was completed by a research assistant (É.K.L). To ensure quality and accuracy, 10% of all prescription entries, 10% of CDEA prescriptions, and 10% of collected patient information from medical records were reviewed by a research coordinator (S.B.). All CDEA recipes, as well as any unclear or atypical prescriptions, were reviewed and confirmed by the hospital pharmacist (D.L.). CDEA classification was discussed by all authors, regarding specific cases, as mentioned in Section 2.3. Patients’ diagnoses and disease classification were determined by a pediatrician (C.L.).

### 2.6. Data Analysis

Data were imported into REDCap and analysed by a biostatistician from the Applied Clinical Research Unit (URCA) of the CHU Sainte-Justine. Results are presented as descriptive statistics. Categorical variables are presented as percentages. For each studied day, the percentage of CDEA and of their various types were calculated based on the total number of drug prescriptions. The proportion of patients receiving at least one CDEA was calculated based on the total number of patients prescribed at least one prescription on the day of interest.

## 3. Results

Of the 727 patients hospitalized on the selected days, a total of 606 (N = 276, summer; N = 330, winter) children (52% male and 48% female) met the inclusion criteria. No differences between seasons were observed, therefore data are presented for the total studied population. Demographic data by age group, distribution across hospital units, and frequency of ICD-11 diagnosis class on admission are described in Table 2. More than half of the total population had already been hospitalized for greater than one week on the selected dates and one third of all their prescriptions were entered into the database on the admission date. Neonates represented the largest age group, accounting for 35% of the entire population. On both days, infectious diseases were the most frequent reason for hospitalization.

As shown in Table 3 and Figure 1, there were 5465 prescriptions registered during both days. Overall, CDEA (n = 714) represented 13% of all prescriptions (enteral and parenteral administration) and 23% of enteral prescriptions only. Across all age groups, the CDEA proportion of enteral prescriptions was greater than 25%, decreasing to 14% for children between 12 and 18 years old. The majority of CDEA (67%) were liquids. Tablets were used in 63% of recipes for suspensions, syrups, or solutions (n = 299). The proportions of liquid CDEA relative to the total number of prescriptions across age groups were higher for younger children, especially for neonates (14%) and for patients under 8 years (11%). Liquid CDEA represented less than 2% (n = 23) of all prescriptions in children above the age of 12 years. Tablet splitting and other solid CDEA accounted for the remaining one third of CDEA. Solid CDEA were more common in patients over the age of 8 years (7%).

Of the 390 prescribed enteral drugs, almost one third required compounding (n = 122). Sixty-seven drugs were compounded into solutions, suspensions, or syrups and 73 CDEA were solid forms. The twenty most prescribed liquids and ten most prescribed solid CDEA in the total population are shown in Figure 2 and Figure 3, respectively. Therapeutic areas for compounded drugs according to the AHFS classification were the central nervous system (35%), cardiovascular (21%), gastro-intestinal (12%), electrolytic, caloric and water balance (9%), hormones and synthetic substitutes (7%), and other drugs (16%). Despite tablet-based liquids being the most prescribed type of CDEA, caffeine as an API-based solution was the most frequent CDEA overall (n = 43), almost exclusively given to neonates. The splitting of lansoprazole orodispersible tablets (ODT) (Prevacid^®^ FasTab) (n = 23) was the most common solid CDEA, with 22 prescriptions for patients under 2 years of age. Lansoprazole was closely followed by acetaminophen tablet splitting (n = 23), exclusively prescribed to children over the age of 8.

Nearly half (N = 298 [49%]) of hospitalized children received at least one prescription for a CDEA, with a median of two CDEA per child (range, 1 to 10). This population received 74% (n = 4018) of all 5465 prescriptions and had a median of 11 prescriptions per child in their medical file (range, 1 to 41). Neonates and teenagers were the groups receiving the least CDEA, 0.9 and 0.8 prescriptions per child, respectively, when compared with others (8 to <12 years, 2.2; 2 to <8 years, 1.7; 28 days to <2 years, 1.2). More than half (N = 49 [65%]) of neonates receiving a CDEA were premature with a gestational age under 32 weeks. The characteristics of patients with at least one prescription for a CDEA are described in Table 4. Hospitalized children with one or more CDEA were overrepresented in multispecialty departments (17 vs. 13%), the rehabilitation centre (15 vs. 9%), the hematology–oncology department (14 vs. 8%), and in the pediatric ICU (12 vs. 7%) when compared with the total population.

## 4. Discussion

This retrospective study highlights how compounding remains a significant part of daily practice for pediatric pharmacotherapy in a hospital setting, representing 23% of all prescriptions for enteral administration and required for 31% of all drugs prescribed during the study period. Furthermore, almost half of hospitalized children received one or more prescriptions requiring compounding across all hospital units. The need for compounding was not isolated to a specific pediatric population and affects any child admitted to the hospital.

It is unsurprising that the majority of CDEA were liquids when prescribed to younger age groups, with solid CDEA being less frequent in this population as swallowing adult size tablets represents a barrier [12,13]. However, interestingly, lansoprazole ODT splitting for children under the age of two was the most common solid CDEA in the overall population. Although lansoprazole is marketed in Canada as an acceptable form for young patients, the available dosage strengths (15 and 30 mg) are inappropriate for children of this age, hence necessitating compounding.

Interestingly, some drugs available as commercial pediatric formulations in Canada, such as acetaminophen, morphine, dimenhydrinate, glycopyrrolate, hydromorphone, and propranolol, required compounding. These represented 21% of all CDEA drugs. In almost half of these cases, the manipulation consisted of splitting tablets for older children. Solid forms may be preferred over commercial liquid formulations in this age group, as the volume per dose for a given concentration may be too large. However, as the available tablet strengths can be inappropriate for the required dose, there could be a need for tablet splitting. Other potential reasons for not administering the commercial forms include reimbursement barriers, caregivers’/healthcare professionals’ preferences for the compounded formulation, or access and supply issues. Acetaminophen is a good example for the need of compounding in pediatrics despite the availability of a commercial child-friendly formulation. In this study, acetaminophen represented 10% of solid CDEA.

Studies evaluating the percentage of prescribed drugs requiring compounding in children are scarce, as published studies have mostly focused on off-label and unlicensed drug use, without providing specific data related to compounding. A 2003 European survey reported that liquid preparations were the extemporaneous preparations of choice in England and Sweden, and solids were selected in France (capsules), Spain (capsules), Finland (powder), and Italy (powder) [14]. The most frequent oral liquid formulations included morphine and caffeine, and the most frequent solids were spironolactone, hydrocortisone, nifedipine, and caffeine. Nevertheless, no conclusions about the practice of a specific country could be drawn. Giam et al. published a systematic review of extemporaneous drugs for pediatric patients in 2008 [15]. Included studies of a retrospective and prospective nature addressed extemporaneous preparations as a subcategory of unlicensed drugs. The percentage of extemporaneous preparations varied from 0 to 42% of prescriptions. The reported limits of this study were the variability of definitions for unlicensed drugs. One study determined the number of claims for compounded drugs on a per user per year basis for all commercially insured patients in the United States during 2012 and 2013 [16]. However, this study was not specific to pediatrics, did not specify the type of compounding or administration route, and was targeted toward community pharmacies. Interestingly, the most frequently compounded drugs in the younger age group (0–10 years), included, among others, lansoprazole, hydrocortisone, baclofen, and spironolactone, concordant with our study’s most prescribed CDEA. Another systematic review on the use of off-label and unlicensed drugs in hospitalized children was published in 2014 and reported that the percentage of unlicensed prescriptions for formulation modifications ranged from 4 to 100% across studies [17]. More recently, a retrospective survey in over two hundred Japanese hospitals was published with a similar research question to our study in patients under the age of 13 years [18]. Approximately 10% of pediatric oral prescriptions required compounding and one third of all 266 compounded API were available as commercial flexible dosage forms. However, the proportion of compounding per age group could not be determined. Two out of the top 10 drugs (hydrocortisone and baclofen) were mutual to this study’s most prescribed CDEA.

Our study has several strengths. To our knowledge, this is the first study evaluating the prevalence of compounded drugs in a pediatric hospital setting in Canada, while also identifying the proportion of children with at least one CDEA and their demographic characteristics. The design addressed potential seasonal variability and the database included an important number of prescriptions (n = 5465) and children (N = 606). Concerning limitations, the study was conducted retrospectively using drug prescription data rather than drug administration data, which could have affected the proportion of CDEA truly received by the patient, as “as needed” prescriptions may have never been given to the patient. The study’s evaluation period included only two days. It was also limited to hospitalized patients in a university-affiliated pediatric hospital (including its rehabilitation centre), which may not accurately reflect the outpatient or the community hospital setting. Additionally, this study did not provide any information regarding compounding of drugs administered parenterally.

Finally, it is concerning to see that six of the top ten (60%) liquid CDEA (domperidone, gabapentin, baclofen, hydrocortisone, levetiracetam, and caffeine) were already marketed as commercial pediatric formulations in the US and/or Europe, for several years in some instances, at the time of the study [9]. To prevent delay in accessing suitable pediatric formulations, the time is now for Health Canada to implement a purposeful regulatory pediatric framework like those of the US and the EU which have proven to be successful for pediatric formulation development and approval. Health Canada’s regulatory reform should include, among others, a pathway with pediatric-specific terms allowing Canadian pediatric submissions to rely on Trusted Foreign Decisions from selected jurisdictions without requesting additional information or data [19].

## 5. Conclusions

The market availability of suitable pediatric formulations in Canada remains challenging, given that compounding was required to treat almost half of the hospitalized pediatric population on two randomly selected days at the second largest Canadian university-affiliated pediatric hospital. Compounding is only one of the many types of off-label uses in the pediatric setting, highlighting the urgency for improving market approval and access to child-friendly pharmaceutical forms. Although the practice of compounding is regulated by provincial pharmacy regulatory authorities and is essential to the provision of drugs to young children, it cannot be considered an equivalent surrogate for a pediatric formulation approved by Health Canada. We can and must do better. Canadian children deserve the same access to pediatric drugs available in suitable child-friendly formulations as children in other countries. International collaboration with major stakeholders such as governments, industry, and academics is essential to facilitate access to pediatric formulations in Canada, as they become available in trusted foreign jurisdictions.

## Figures and Tables

**Figure 1 children-10-00147-f001:**
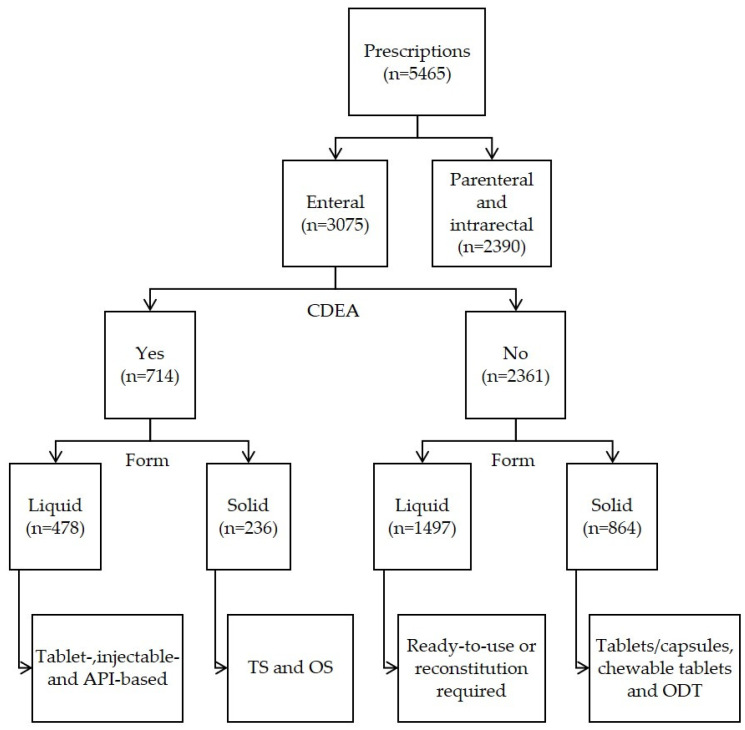
Flow chart of prescription type identification results. Abbreviations: API, active pharmaceutical ingredient; CDEA, compounded drug for enteral administration; ODT, orodispersible tablet; OS, other solid; TS, tablet splitting.

**Figure 2 children-10-00147-f002:**
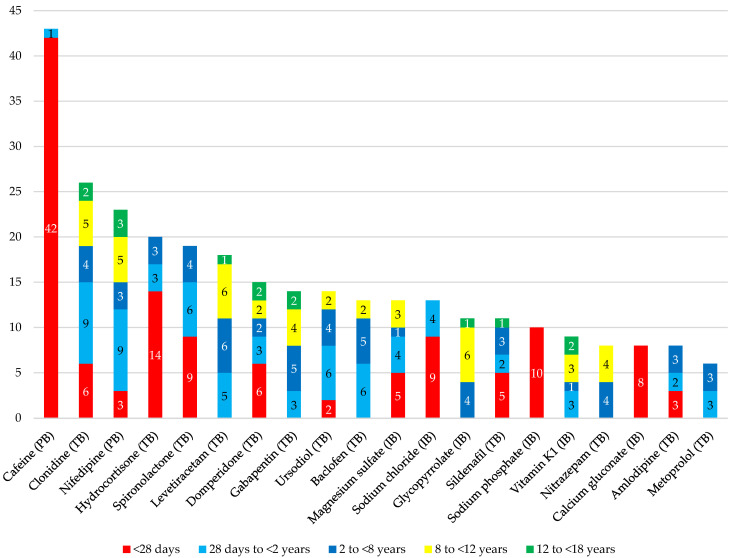
Twenty most prescribed liquid CDEA in the total population. Abbreviations: IB, injectable-based oral solution; PB, powder-based solution, suspension, or syrup; TB, tablet-based solution, suspension, or syrup.

**Figure 3 children-10-00147-f003:**
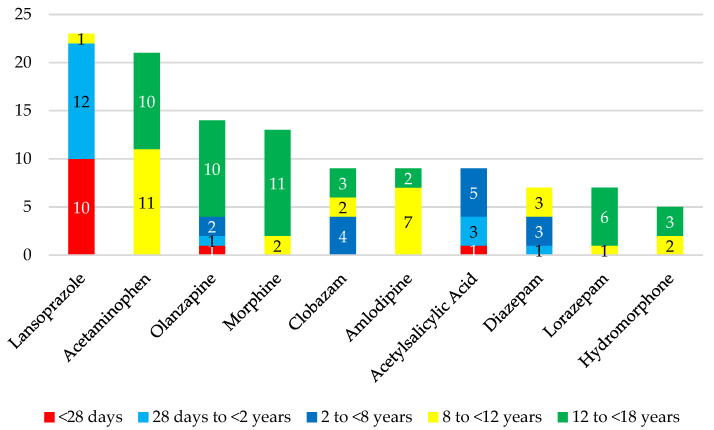
Ten most prescribed solid CDEA in the total population.

**Table 1 children-10-00147-t001:** Prescription definitions for enteral drugs.

Prescription Type	Description	CDEA
Tablet-or capsule based liquid	Tablet typically crushed into a fine powder or content of capsule dispersed before diluting or mixing into syrup or other liquid	Yes
Injectable-based liquid	Injectable drug diluted for oral administration or without dilution	Yes
API-based liquid	API directly diluted or mixed into syrup or other liquid	Yes
Tablet Splitting	Tablet scored and unscored tablets administered in fractions of dosage	Yes
Other solid	May include API encapsulated for appropriate pediatric dosing or a marketed capsule opened to weigh and split API (powder or granules)	Yes
Ready-to-use	Marketed as is	No
Reconstitution required	Water dilution of bottled API required as indicated by the manufacturer	No
Other liquid	Injectable liquid drug administered orally (no dilution required)	No
Tablets/capsules	Marketed as is	No
Chewable tablets	Chewing required before swallowing	No
ODT	No chewing or drinking liquids needed to swallow and ingest, as API disintegrates rapidly in saliva	No
Other solid	Marketed as gums, throat lozenges, or dispersible tablets	No

Abbreviations: API, active pharmaceutical ingredient; CDEA, compounded drug for enteral administration; ODT, orodispersible tablet.

**Table 2 children-10-00147-t002:** Demographic data of the total population per age group (N = 606).

		Age Group				
Parameter	Total (N = 606)	<28 Days (N = 213)	28 Days to <2 Years (N = 108)	2 to <8 Years (N = 85)	8 to <12 Years (N = 71)	12 to <18 Years (N = 129)
Sex, N (%)						
Male	318 (52)	119 (56)	65 (60)	47 (55)	38 (54)	49 (38)
Hospital unit, N (%)						
Pediatrics	124 (20)	11 (5)	40 (37)	22 (26)	15 (21)	36 (28)
NICU ^1^	121 (20)	119 (56)	2 (2)			
Multispecialty	78 (13)	8 (4)	16 (15)	11 (13)	9 (13)	34 (26)
Surgery	74 (12)	5 (2)	18 (17)	16 (19)	12 (17)	23 (18)
Nursery ^2^	56 (9)	56 (26)				
Rehabilitation Centre	56 (9)	1 (0)	4 (4)	18 (21)	18 (25)	15 (12)
Hematology-Oncology	50 (8)		10 (9)	12 (14)	11 (15)	17 (13)
Pediatric ICU	47 (8)	13 (6)	18 (17)	6 (7)	6 (8)	4 (3)
ICD-11 class of diagnosis, N (%)						
Infectious Diseases	97 (16)	12 (6)	41 (38)	28 (33)	10 (14)	6 (5)
Preterm Birth	92 (15)	91 (43)	1 (1)			
Term Birth	62 (10)	62 (29)				
Hematology-Oncology	62 (10)		10 (9)	14 (16)	18 (25)	20 (16)
Neurology	55 (9)	4 (2)	3 (3)	17 (20)	18 (25)	13 (10)
Digestive System	52 (9)	10 (5)	15 (14)	7 (8)	6 (8)	14 (11)
Psychiatry	44 (7)			1 (1)	1 (1)	42 (33)
Cardiovascular System	30 (5)	13 (6)	10 (9)	5 (6)		2 (2)
Trauma	30 (5)		5 (5)	6 (7)	8 (11)	11 (9)
Others ^3^	82 (13)	21 (10)	23 (21)	7 (8)	10 (14)	21 (16)

Abbreviations: ICD-11, the 11th International Classification of Diseases; ICU, intensive care unit; NICU, neonatal intensive care unit. ^1^ Median gestational age was 30 weeks for preterm neonates and 39 weeks for term neonates in the NICU. ^2^ Median gestational age was 36 weeks for preterm neonates and 39^3/7^ weeks for term neonates in the nursery. ^3^ Others include dermatology, endocrine system, genetics, immune system, metabolic disorders, nephrology and urogenital system, non-accidental trauma, orthopedics, otorhinolaryngology and respiratory system.

**Table 3 children-10-00147-t003:** Active prescriptions for the total population (N = 606).

	Age Group					
Prescription Type	Total (N = 5465)	<28 Days (N = 1225)	28 Days to <2 Years (N = 893)	2 to <8 Years (N = 1000)	8 to <12 Years (N = 1054)	12 to <18 Years (N = 1293)
Enteral administration, N(%) ^1^	3075 (56)	712 (58)	454 (51)	550 (55)	584 (55)	775 (60)
Commercial form	2361 (43)	530 (43)	327 (37)	409 (41)	428 (41)	667 (52)
CDEA	714 (13)	182 (15)	127 (14)	141 (14)	156 (15)	108 (8)
🢖Liquid	478 (9)	168 (14)	96 (11)	106 (11)	85 (8)	23 (2)
🢒Tablet/capsule-based	299 (5)	74 (6)	68 (8)	91 (9)	53 (5)	13 (1)
🢒Injectable-based	92 (2)	38 (3)	13 (1)	9 (1)	25 (2)	7 (1)
🢒API-based	87 (2)	56 (5)	15 (2)	6 (1)	7 (1)	3 (0)
🢖Solid	236 (4)	14 (1)	31 (3)	35 (3)	71 (7)	85 (7)
🢒Tablet splitting	231 (4)	13 (1)	28 (3)	35 (3)	70 (7)	85 (7)
🢒Others ^2^	5 (0)	1 (0)	3 (0)		1 (0)	
Parenteral administration. N(%) ^1^	2390 (44)	513 (42)	439 (49)	450 (45)	470 (45)	518 (40)

Abbreviations: API, active pharmaceutical ingredient; CDEA, compounded drug for enteral administration. ^1^ Percentage represents the proportion of prescriptions on total active prescriptions for the total population and per age group. ^2^ Others include powder measurement, capsule opening and API spreading, and API encapsulation.

**Table 4 children-10-00147-t004:** Patients receiving at least one CDEA (N = 298).

		Age Group				
Parameter	Total (N = 298)	<28 Days (N = 75)	28 Days to <2 Years (N = 55)	2 to <8 Years (N = 51)	8 to <12 Years (N = 51)	12 to <18 Years (N = 66)
Sex, N (%)						
Male	160 (54)	41 (55)	35 (64)	29 (57)	27 (53)	28 (42)
Hospital Unit, N (%)						
Pediatric ICU	37 (12)	10 (13)	13 (24)	5 (9)	6 (12)	3 (5)
NICU	58 (19)	58 (77)				
Hematology–Oncology	43 (14)		8 (15)	11 (22)	10 (20)	14 (21)
Pediatrics	40 (13)	1 (1)	13 (24)	8 (16)	6 (12)	12 (18)
Surgery	25 (8)		6 (11)	3 (6)	7 (14)	9 (14)
Multispecialty	50 (17)	5 (7)	11 (20)	8 (16)	7 (14)	19 (29)
Rehabilitation Centre	45 (15)	1 (1)	4 (7)	16 (31)	15 (29)	9 (14)

Abbreviations: ICU, intensive care unit; NICU, neonatal intensive care unit.

## Data Availability

Data and materials are available upon request to the corresponding author.

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
