# Peer review of "Drug Prescriptions Requiring Compounding at a Canadian University Affiliated Pediatric Hospital: A Cross-Sectional Study"

_children, 2023, doi:10.3390/children10010147_

Round 1

Reviewer 1 Report

This is a cross-sections study of the frequency and types of compounded nonsterile preparations prescribed for children at one Canadian pediatric hospital at two randomly selected recent time points. The manuscript is well written and adds to the literature in this important area. The authors raise a number of important questions, especially related to the lack of available approved products in Canada and the compounding of those already available, like acetaminophen. The title and abstract are descriptive of the work. The limitations are well explained. Conclusions follow from the results. 

My only criticism is that the references are not in MDPI style and reference 5 is incomplete.

Author Response

Please see attachement.

Reviewer 2 Report

Many thanks to the authors who worked hard on this article. I would like to thank the MDPI editors who invited me to review this manuscript which aimed to determine the proportion of prescriptions 54 for compounded drugs for enteral administration (CDEA), among all enteral and 55 parenteral prescriptions, in children under the age of 18 years hospitalized at the CHU 56 Sainte-Justine and its rehabilitation center, a Canadian university-affiliated 484-bed 57 tertiary care pediatric hospital in Montreal, Quebec and to determine the proportion of prescribed drugs requiring compounding and the proportion 59 of children prescribed at least one CDEA during their hospital stay.

In my opinion, the section introduction should be added

Materials and Methods section should be more clarified
